# Development and Optimization of a Selective Whole-Genome Amplification To Study *Plasmodium ovale* Spp.

V. Joste,[a,b,c] E. Guillochon,[a] J. Clain,[a] R. Coppée,[a]* S. Houzé[a,b,c]

ᵃUniversité Paris Cité, IRD, MERIT, Paris, France
ᵇCentre National de Référence du Paludisme, AP-HP, Hôpital Bichat-Claude-Bernard, Paris, France
ᶜLaboratoire de parasitologie-mycologie, AP-HP, Hôpital Bichat-Claude-Bernard, Paris, France

**ABSTRACT** Since 2010, the human-infecting malaria parasite *Plasmodium ovale* spp. has been divided into two genetically distinct species, *P. ovale wallikeri* and *P. ovale curtisi*. In recent years, application of whole-genome sequencing (WGS) to *P. ovale* spp. allowed to get a better understanding of its evolutionary history and discover some specific genetic patterns. Nevertheless, WGS data from *P. ovale* spp. are still scarce due to several drawbacks, including a high level of human DNA contamination in blood samples, infections with commonly low parasite density, and the lack of robust *in vitro* culture. Here, we developed two selective whole-genome amplification (sWGA) protocols that were tested on six *P. ovale wallikeri* and five *P. ovale curtisi* mono-infection clinical samples. Blood leukodepletion by a cellulose-based filtration was used as the gold standard for intraspecies comparative genomics with sWGA. We also demonstrated the importance of genomic DNA preincubation with the endonuclease McrBC to optimize *P. ovale* spp. sWGA. We obtained high-quality WGS data with more than 80% of the genome covered by ≥5 reads for each sample and identified more than 5,000 unique single-nucleotide polymorphisms (SNPs) per species. We also identified some amino acid changes in *pocdhfr* and *powdhfr* for which similar mutations in *P. falciparum* and *P. vivax* are associated with pyrimethamine or cycloguanil resistance. In conclusion, we developed two sWGA protocols for *P. ovale* spp. WGS that will help to design much-needed large-scale *P. ovale* spp. population studies.

**IMPORTANCE** *Plasmodium ovale* spp. has the ability to cause relapse, defined as recurring asexual parasitemia originating from liver-dormant forms. Whole-genome sequencing (WGS) data are of importance to identify putative molecular markers associated with relapse or other virulence mechanisms. Due to low parasitemia encountered in *P. ovale* spp. infections and no *in vitro* culture available, WGS of *P. ovale* spp. is challenging. Blood leukodepletion by filtration has been used, but no technique exists yet to increase the quantity of parasite DNA over human DNA when starting from genomic DNA extracted from whole blood. Here, we demonstrated that selective whole-genome amplification (sWGA) is an easy-to-use protocol to obtain high-quality WGS data for both *P. ovale* spp. species from unprocessed blood samples. The new method will facilitate *P. ovale* spp. population genomic studies.

**KEYWORDS** dihydrofolate reductase, Mcrbc endonuclease, *Plasmodium ovale* spp., orthologs, sWGA, whole genome

Address correspondence to V. Joste, valentinjoste@gmail.com.

*Present address: R. Coppée, Université Paris Cité & Sorbonne Paris Nord, Inserm, IAME, Paris, France.

The authors declare no conflict of interest.

**M**alaria is a vector-borne infectious disease transmitted by *Anopheles* mosquito bites. The main agent of human malaria in terms of number of clinical cases and related deaths is *Plasmodium falciparum* (1). *P. ovale* spp. represents 0.77% of *Plasmodium* cases worldwide and up to 1.69% in Africa (2). According to the French National Malaria Reference Center (FNMRC) data, *P. ovale* spp. is the second most frequently detected malaria species among imported malaria cases during the last decade and represented

from 6% to 10% of total malaria infections in France (3). Since 2010, *P. ovale* spp. has been divided into two genetically distinct species, *P. ovale wallikeri* and *P. ovale curtisi* (4). These two species are sympatric in Africa and differ by their clinical and biological characteristics in infected travelers (5 to 8).

Like *P. vivax*, *P. ovale* spp. parasites have the ability to cause relapse, defined as recurring asexual parasitemia originating from liver-dormant forms (9). The first reported cases of *P. ovale* spp. relapses date back to 1955 (10). A retrospective study on patients treated for neurosyphilis with sporozoite and trophozoite-induced *P. ovale* spp. malaria infections (also called malaria therapy) revealed that half of them had a relapse event after a complete chloroquine course (11). The existence of hypnozoites in *P. ovale* spp. is still debated (9), but Soulard et al. recently noticed late-developing schizonts in humanized mice infected with *P. ovale* spp. sporozoites (12).

The study of *P. ovale wallikeri* and *P. ovale curtisi* genomes could provide a better understanding of the genomic diversity and population structures of the two species, and help to date the species separation event and identify putative molecular markers associated with the relapse mechanism. In 2017, the first assembled genomes of *P. ovale wallikeri* and *P. ovale curtisi* were published, allowing to branch with confidence these species within the *Plasmodium* phylogeny (13). Of note, the study reported *P. ovale curtisi* genes orthologous to *P. falciparum* and *P. vivax*, but *P. ovale wallikeri* orthologs have not yet been published (13), limiting orthologous-based comparisons at both the *Plasmodium* and *P. ovale* spp. scales.

Whole-genome sequencing (WGS) of *P. ovale* spp. remains challenging due to the low parasite density commonly found in *P. ovale* spp. infections, compared to *P. falciparum* and *P. vivax*, and the absence of *ex vivo* culture protocols to amplify the parasites *in vitro* (14). In addition, parasite DNA extracted from unprocessed whole blood is highly contaminated with human DNA, requiring parasite DNA isolation or selective parasite genome amplification before sequencing. Some techniques have been developed to enrich the *Plasmodium* genome from clinical blood samples by filtering out leukocytes carrying human DNA before DNA extraction (15), selectively amplifying the parasite genome using specific primer sets, referred to as the selective whole-genome amplification (sWGA) (16) or hybrid selection (17). The sWGA approach has already been successfully developed for *P. falciparum* (18), *P. vivax* (19), *P. malariae* (20), and *P. knowlesi* (21) human-infecting malaria parasites.

In this work, we developed specific sWGA protocols for *P. ovale wallikeri* and *P. ovale curtisi*. We showed that this method is efficient and cost-effective to obtain high-quality *P. ovale* spp. genomic data, with blood leukodepletion by filtration used as the comparative gold standard (22). Digestion of genomic DNA with the restriction enzyme McrBC, an endonuclease that cleaves DNA containing methylcytosine (23), improved sWGA quality, as previously observed for low-density *P. falciparum* samples (24). Through the 11 new *P. ovale* spp. genomes sequenced using this method, we identified more than 5,000 single-nucleotide polymorphisms (SNPs) in both species, and we assigned *P. ovale wallikeri* genes orthologous to *P. ovale curtisi*, based on the previously published *P. ovale* spp. assemblies (13).

## RESULTS

**Sample collection.** A total of five *P. ovale curtisi* (here named Poc1 to Poc5) and six *P. ovale wallikeri* (Pow1 to Pow6) isolates were included in the study, covering a wide range of parasite density (*P. ovale curtisi*: 1,790 to 26,700 parasites/$\mu$L [p/$\mu$L] and cycle threshold (Ct) 23.2 to 28.1; *P. ovale wallikeri*:198 to 90,000 p/$\mu$L and Ct 23 to 31.2) (Table 1). Isolates were originated from patients who travelled to West or Central Africa. All samples were used to develop the sWGA strategy, with or without McrBC digestion. Poc1 and Pow1 were selected to perform leukodepletion as positive controls for high-quality WGS.

**Custom-made cellulose-based filtration is a suitable leukodepletion method for *P. ovale* spp. clinical samples.** In order to evaluate sWGA in producing accurate WGS data, the leukodepletion-based method was used as the gold standard (15), as it

**TABLE 1** Isolates included in the study[a]

| Sample | Date of inclusion | Country of contamination | Parasite density (p/µL) | Ct *P. ovale* spp. | Ct human | Delta | sWGA | sWGA + McrBc | Filtration | Chromosomes SNPs filtered (sWGA/filtration) |
|---|---|---|---|---|---|---|---|---|---|---|
| Poc1 | April 2021 | Republic of the Congo | 8,000 | 25 | 21 | 4 | X | X | X | 3,732/6,980 |
| Poc2 | January 2019 | Gabon | 8,000 | 24, 1 | 20, 9 | 3, 2 | X | X | | 3,893/NA |
| Poc3 | March 2015 | Central African Republic | 20,000 | 28, 1 | 21, 9 | 6, 2 | X | X | | 4,823/NA |
| Poc4 | November 2016 | Cameroon | 26,700 | 23, 2 | 20, 3 | 2, 9 | X | X | | 2,558/NA |
| Poc5 | September 2018 | Ivory Coast | 1,790 | 27 | 21, 3 | 5,7 | X | X | | 1,580/NA |
| Pow1 | April 2021 | Cameroon | 22,000 | 24 | 20 | 4 | X | X | X | 6,045/6,145 |
| Pow2 | October 2018 | Mali | 90,000 | 23 | 22, 4 | 0,6 | X | X | | 5,292/NA |
| Pow3 | December 2015 | Benin | 198 | 31, 2 | 23, 6 | 7, 6 | X | X | | 5,430/NA |
| Pow4 | April 2018 | Cameroon | 2,961 | 28 | 21, 5 | 6, 5 | X | X | | 5,055/NA |
| Pow5 | December 2019 | Central African Republic | 8,000 | 25 | 21 | 4 | X | X | | 6,402/NA |
| Pow6 | August 2018 | Republic of the Congo | 450 | 29, 8 | 22 | 7, 8 | X | X | | 4,864/NA |

[a]Poc1 to Poc5 and Pow1 to Pow6 were tested on both sWGA and sWGA + McrBc conditions. Poc1 and Pow1 were also tested for the filtration condition. qPCR cycle thresholds (Ct) have been obtained with the *Plasmodium* typage kit (see Materials and Methods). Delta is the difference between the Ct of *P. ovale* spp. and the Ct of the human. Chromosome SNPs filtered column indicates the number of SNPs obtained on the reconstructed chromosome for the sWGA + McrBc condition. A "X" symbol indicates that the experiment has been performed. NA, not applicable.

was already successfully used for *P. ovale* spp. genome sequencing (13). When applied to Poc1 and Pow1 red blood cells, filtration successfully removed human leukocytes, with a 1,000- to 10,000-fold loss of human template DNA as quantified by qPCR targeting the human *beta actin* gene, whereas parasite DNA loss was very much less (Table S1 in the supplemental material). WGS of those samples revealed that most of the reads mapped to the parasite genome (74% for Poc1, 95% for Pow1). More than 95% of the parasite genome was covered with a depth of coverage ≥10×.

**Newly developed sWGA primers and McrBC preincubation enrich *P. ovale* spp. DNA over human DNA.** Despite blood leukodepletion being a suitable method for getting high-quality *P. ovale* spp. genomes, it is time-demanding, especially for large-scale biobanking, and cannot be used when the available blood volume is low. To fill this gap, we aimed to develop an sWGA approach for both *P. ovale curtisi* and *P. ovale wallikeri* that can be applied to archived DNA biobanks from unfiltered clinical blood samples.

Primers' sets with phosphorothioate bonds (*) that reached the best score consisted of five primers for both *P. ovale curtisi* (Poc set: ATATTTT*C*G, CGTAT*C*G, TAATTCG*T*A, TATTTCG*T*A, and TCGTATA*T*A) and *P. ovale wallikeri* (Pow set: ATATACG*A*A, CGA TAAA*A*A, CGATA*C*G, TACGAAA*T*A, and TATAACG*A*A). For the Poc set, the mean distances between two primers on human and *P. ovale curtisi* genomes were respectively 128,170 and 6,964 nucleotides. For the Pow set, the mean distances between two primers on human and *P. ovale wallikeri* genomes were respectively 115,670 and 6,082 nucleotides. Number of primers' hits per genome was significantly higher in *P. ovale wallikeri* compared to *P. ovale curtisi* (median [10th percentile to 90th percentile]: 4,551 [4,479 to 4,628] versus 4986 [4934 to 5003]; $P = 0.004$, Mann-Whitney $U$ test) (Table S2).

After WGS, mean depth of coverage with sWGA for the five *P. ovale curtisi* and the six *P. ovale wallikeri* was respectively 32× and 24×. For *P. ovale curtisi*, approximately half of the reads mapped to the parasite genome, whereas one third only mapped to the parasite genome for *P. ovale wallikeri*. For both *P. ovale curtisi* and *P. ovale wallikeri*, half of their genome was covered at ≥10×, suggesting that some improvements in the method could be done (Table S3a).

In a previous study, it was demonstrated that digesting the genomic DNA with McrBC endonuclease enzyme prior to sWGA greatly improved WGS data for low *P. falciparum* parasite densities (24). Considering that *P. ovale* spp. samples were commonly associated with low parasite density, we decided to explore the effect of McrBC digestion prior to sWGA on WGS results. WGS of McrBC-treated samples revealed a significant improvement in data quality, with a rise in the percentage of reads mapping to the parasite genome ($P < 0.001$ for both species, $\chi^2$ test), in the normalized mean coverage ($P = 0.002$; Mann-Whitney $U$ test) and in the percentage of the genome with a depth of coverage ≥10× ($P < 0.001$, $\chi^2$ test) (Fig. 1A and B and Table S3A). For each sample, each chromosome had a mean coverage ≥10× for both species (Fig. 2A and B) except for the chromosome 4 of *P. ovale curtisi*. With the McrBC enzyme, *P. ovale wallikeri* and *P. ovale curtisi* had equivalent normalized mean coverage ($P = 0.66$, Mann-Whitney $U$ test), but *P. ovale wallikeri* displayed a higher percentage of genome covered with at least 10 reads ($P < 0.001$, $\chi^2$ test). The only sample for which the McrBC enzyme did not improve the overall WGS data quality was the sample Pow2 (Table S3A and S3B and Fig. S1). This sample was associated with the highest parasite density, the lowest qPCR Ct for *P. ovale wallikeri*, and the lowest ΔCt (Table 1). Hereafter, McrBC condition was chosen for downstream analyses. The specificity of the short reads generated by sWGA was confirmed by aligning the raw reads of Poc1 and Pow1 with the *P. falciparum* and *P. malariae* reference genomes (Fig. S2).

We finally compared WGS data between leukodepletion and McrBC-treated sWGA approaches. Normalized mean coverages were similar between the filtration and the McrBC conditions and equivalent percentage of reads mapped to the *P. ovale* spp. genomes. In contrast, depth of coverage at ≥10× was significantly better by using filtration (Fig. S3) in concordance with a better homogeneity in reads mapping (Fig. 3), as already reported for other *Plasmodium* species (18, 25).

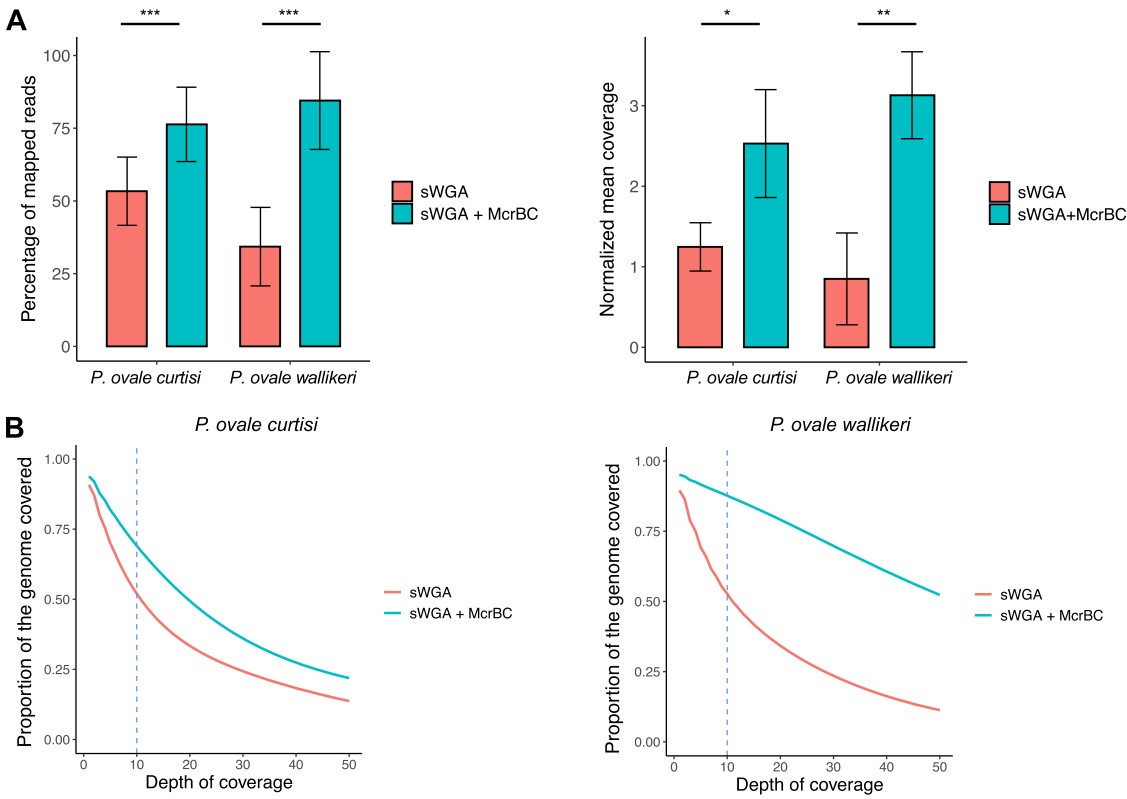

**FIG 1** (A) Percentage of mapped reads (left) and normalized mean coverage (right) for sWGA in red and sWGA with preincubation with the McrBC enzyme in blue. Normalized mean coverage is the mean coverage divided by the number of total reads produced. (B) Proportion of the genome covered with or without the McrBC endonuclease enzyme for both *P. ovale curtisi* (left) and *P. ovale wallikeri* (right). The blue dashed line represents a depth of coverage of 10. *, $P < 0.05$; **, $P < 0.01$; ***, $P < 0.001$.

**Orthologous genes and interspecies phylogenetic tree.** To perform comparative genomics at both the *Plasmodium* and *P. ovale* spp. scales, we first determined *P. ovale wallikeri* orthologous protein-coding genes that were still to be characterized (13). We first excluded the 1,742 *pir* proteins from the PocGH01 reference proteome (13), which are encoded by a multigenic family with numerous amino acid sequence variations, similarly to *var* genes in *P. falciparum* (26). By aligning *P. ovale curtisi* on *P. ovale walli-keri* protein sequences with BLAST+ (27), we identified 4,420 proteins with a high sequence similarity. We validated the orthology with OrthoMCL (28) for 4,175 proteins (3,626 orthologs between *P. ovale wallikeri* and *P. falciparum*, 3,776 orthologs between

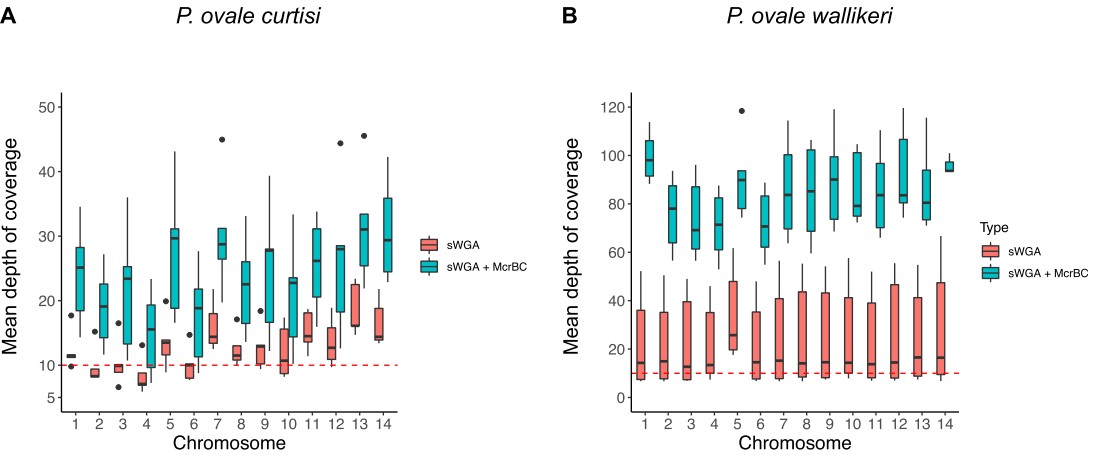

**FIG 2** Mean depth of coverage on each chromosome *P. ovale curtisi* (A) and *P. ovale wallikeri* (B) with sWGA (in red) or sWGA with preincubation with the McrBC enzyme (in blue). The red dashed line represents a depth of coverage of 10.

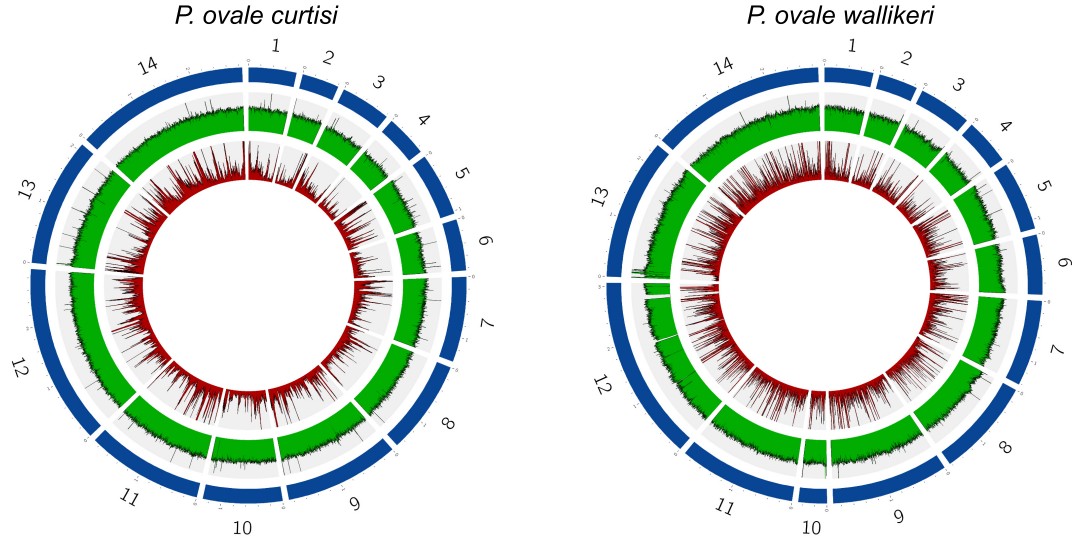

**FIG 3** Comparison of the reads mapping homogeneity for the sWGA + McrBC condition (in red) and the filtration condition (in green) among the 14 chromosomes of *P. ovale curtisi* (on the left) and *P. ovale wallikeri* (on the right).

*P. ovale wallikeri* and *P. vivax*). We found 121 *P. ovale curtisi* and *P. ovale wallikeri* proteins in distinct OrthoMCL ortholog groups, and 124 *P. ovale wallikeri* proteins did not belong to any ortholog group (Table S4).

Using a subset of 216 orthologous protein sequences across 13 *Plasmodium* species and including our 11 *P. ovale* spp. samples, a phylogenetic tree was built (Table S5). The phylogenetic relationships obtained with this set of orthologous sequences were largely consistent with the acknowledged phylogeny of *Plasmodium* species (Fig. 4).

**Identification of SNPs in *P. ovale curtisi* and *P. ovale wallikeri* clinical samples.** To identify SNPs from sWGA reads, we performed a variant calling based on the reconstructed chromosomes (excluding the unassigned contigs). From sWGA-treated samples, a mean of 3,317 per sample and 5,515 SNPs per sample were identified in the chromosomes of the five *P. ovale curtisi* and the six *P. ovale wallikeri* isolates, respectively. No evidence of multiclonal isolates was detected according to the nonreference allele frequency (NRAF) plots (Fig. S4A and S4B).

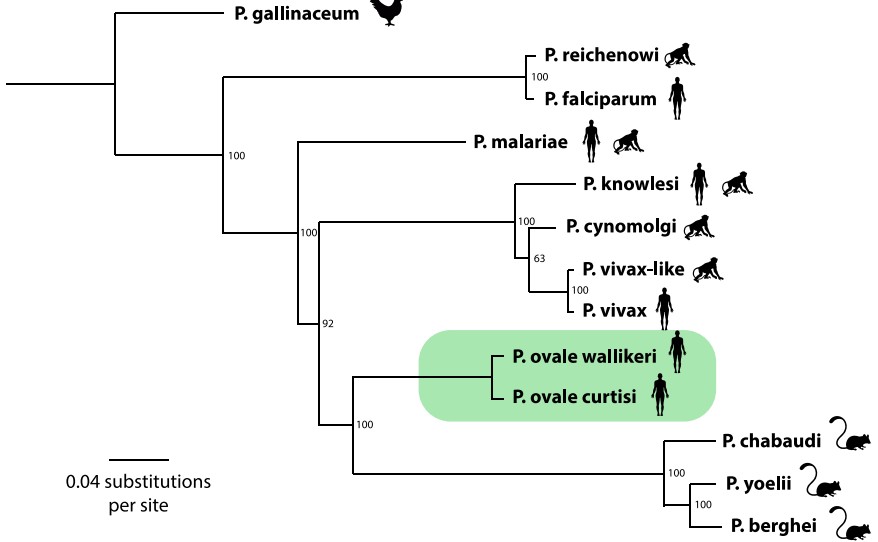

**FIG 4** Maximum likelihood phylogenetic tree using protein sequences of the *Plasmodium* genus rooted on *P. gallinaceum* showing the *P. ovale* spp. clade in green. Bootstrap values are indicated at nodes. Pictures indicate the host specificities.

**TABLE 2** Amino acid changes in *P. ovale* spp. orthologous proteins to major *P. falciparum* drug-resistance genes

| Species | Isolate | Crt | Dhfr | Dhps | Mdr | k13 |
|---|---|---|---|---|---|---|
| *P. ovale curtisi* | Poc1 | | H98P | K189E and D275G | I1303V | |
| | Poc2 | | H98P | | I1303V | |
| | Poc3 | | A15S, S58R and H98P | | | |
| | Poc4 | | A15S | | | |
| | Poc5 | | | | | |
| *P. ovale wallikeri* | Pow1 | C19G | | | F34N | |
| | Pow2 | C19G | | | | |
| | Pow3 | C19G | | | | |
| | Pow4 | C19G and L216F | | | | |
| | Pow5 | | F57L and S58R | | | |
| | Pow6 | C19G and L216F | S113N | | | |

To confirm the accuracy of sWGA, we compared the NRAF metrics for the positions with SNPs (NRAF > 0) for samples subjected to filtration and sWGA approaches (i.e., Poc1 and Pow1). We called 3,732 and 6,980 SNPs in sWGA and leukodepletion for *P. ovale curtisi* and 6,045 and 6,145 for *P. ovale wallikeri*. We obtained highly correlated NRAF for both species ($\rho$ = 0.75 for *P. ovale curtisi*, $\rho$ = 0.87 for *P. ovale wallikeri*; $P$ < 0.001, Spearman's rank correlation test, Fig. S5), indicating that filtration and sWGA in the context of *P. ovale* spp. led to very similar variant calling. Most of the differences between the two methods were due to coverage or quality issues (Fig. S6A and S6B). Positions that had different genotypes (i.e., a wild-type allele in one method and heterozygous or homozygous mutant allele at the same position with the other method) were mainly in noncoding regions (62% of discordant positions for *P. ovale curtisi* and 68% for *P. ovale wallikeri*) and with low NRAF (Fig. S6C).

We finally looked specifically for SNPs in *P. ovale* spp. genes orthologous to major *P. falciparum* drug resistance genes. SNPs were detected in the intron and/or coding sequence of the chloroquine resistance transporter (*powcrt* and *poccrt*), dihydrofolate reductase (*pocdhfr* and *powdhfr*), dihydropteroate synthetase (*pocdhps* and *powdhps*), and multidrug resistance protein 1 (*pocmdr1* or *powmdr1*) genes (Table S6). No mutation was detected in the *kelch13* genes (*pock13* and *powk13*). Amino acid changes were found in *pocdhfr* (A15S and S58R), *powdhfr* (F57L, S58R, S113N), *pocdhps* (K189E, D275G), *powcrt* (C19G, L216F), and *powmdr1* (F34N) (Table 2). Remarkably, the mutations *pocdhfr* A15S, *pocdhfr* S58R, *powdhfr* S58R, and *powdhfr* S113N corresponded to positions known to confer pyrimethamine and cycloguanil resistance in *P. falciparum* (*pfdhfr* A16V, C59R, and S108N) (29, 30) (Fig. S7). The mutation *powdhfr* F57L corresponded to the *pvdhfr* F57L/I known to confer pyrimethamine resistance in *P. vivax* (31). Reads from previous *P. falciparum* and *P. malariae* WGS (ERR636035 for *P. falciparum* and ERR4019168 for *P. malariae*) did not map to those *P. ovale* spp. orthologous genes, confirming the specificity of the SNPs generated (Fig. S8).

## DISCUSSION

*P. ovale* spp. was divided in 2010 into *P. ovale curtisi* and *P. ovale wallikeri* based on distinct genetic patterns, absence of hybrid forms despite sympatry, and potential incompatibility of sexual-encoding proteins (4, 8, 31). The two *P. ovale* spp. have been precisely studied in terms of epidemiology, diagnostics, biology, and epidemiology, based on imported malaria cases (5 to 8). However, very few genomic data are available and molecular evolutionary analysis remains incomplete for these species.

We aimed to develop sWGA for both *P. ovale curtisi* and *P. ovale wallikeri* to sequence the parasite genome from archived DNA that had not been filtered. Specific *P. ovale* spp. DNA amplification through sWGA with McrBC preincubation is a suitable method to obtain high-quality sequences and SNPs for variant analysis, within the parasite density tested (up to 1,790 p/$\mu$L for *P. ovale curtisi* and 198 p/$\mu$L for *P. ovale wallikeri*). In this study, we identified an average of 3,317 and 5,515 SNPs per sample in the chromosomes of *P. ovale curtisi* and *P. ovale wallikeri*, respectively, less than what was

previously described in *P. vivax* (14,000 per sample in a study on 18 samples [19]) and *P. malariae* (5,800 per sample in a study on 18 samples [20]). Due to imperfect assembly, only 60% of the *P. ovale* spp. genomes are assembled into chromosomes, and the rest of the genomes were grouped in nearly 700 contigs (13). As contigs contain hypervariable regions like *pir* genes, SNPs were called only on chromosomes, probably explaining the lower number of SNPs compared to *P. malariae* or *P. vivax*, since the core genome (i.e., the genome region used for variant calling analysis) of these species represents 80% and 92% of their entire genome, respectively (20, 32). Although the NRAF plot did not show evidence of polyclonal isolates, some positions were heterozygous, especially for *P. ovale wallikeri* (Fig. S4). Those features might relate to sequencing errors or alignment artifacts such as incorrect mapping of reads due to paralogous regions (32), described by Rutledge et al. (13). Better reconstructions of the two *P. ovale* spp. genomes and more genomic data will probably help in the future to improve variant filtering, such as previously described by Pearson et al. for *P. vivax* (32), and to reduce the gap in the SNP number with the other human-infecting *Plasmodium* species. Finally, to ensure that sWGA on *P. ovale* spp. provides correct sequencing data, we compared the SNPs obtained from this strategy with those called from the same samples but processed in parallel with the blood-filtration strategy. NRAF estimated from these paired samples was strongly correlated for both species (Fig. S5). We missed some SNPs in sWGA mainly due to insufficient coverage related to nonhomogeneity of the reads mapping (Fig. 3).

We further demonstrated the efficiency of sWGA to amplify orthologous genes to those associated with drug resistance in *P. falciparum* or *P. vivax* and identify SNPs in *powcrt*, *pocdhfr*, *powdhfr*, *pocdhps*, *pocmdr1*, and *powmdr1*. Some *P. ovale* spp. *dhfr* mutations led to amino acid changes at positions known to reduce susceptibility to pyrimethamine and cycloguanil in *P. falciparum* or *P. vivax* (A15S, F57L, S58R, and S113N; see Fig. S7) (31, 33). The biological effect, the prevalence, and the geographical origin of these mutations are unknown in *P. ovale* spp., and both molecular epidemiology and *in vitro* testing studies should be performed. Of note, a previous study reported amino acid change S113C in two *P. ovale curtisi* isolates from the China-Myanmar border area (34), and in another study, three *P. ovale curtisi* with A15S + S58R and one *P. ovale wallikeri* with F57L + S58R in Africa (35).

Although sWGA could be of great help to perform *P. ovale* spp. genomic study, it presents two major limitations. First, amplification of *Plasmodium* coinfections is really challenging. Even though theoretically possible, sequencing of *P. ovale* spp. in coinfection with *P. falciparum* (more frequent in areas of endemicity [36] than in imported malaria [37]) might be difficult. While *P. ovale* spp. sWGA primers bind more frequently to *P. ovale* spp. than to *P. falciparum* genome (Table S2), differences in parasite densities (10 times higher for *P. falciparum* than *P. ovale* spp. [38]) remain a major obstacle for both sWGA and leukodepletion. Capture methods to selectively collect one *Plasmodium* species DNA in a mixed infection should be further explored (17, 39). Second, due to preamplification with Phi29 polymerase resulting in nonhomogeneity of reads mapping, gene copy number variation (40) could not be studied with sWGA, despite its importance in *Plasmodium* resistance to treatments (41).

Genomic analysis of *P. ovale curtisi* and *P. ovale wallikeri* is still in its infancy. There is a need to better understand the differences between the two species, including a dating of the separation event, and, more importantly, to identify molecular determinant(s) of medically relevant traits such as dormancy. Further studies, with a larger number of isolates from different geographic areas, are needed to characterize the genetic diversity of *P. ovale* spp., with the potential to discover features that will help to control the disease. The new tools and genome data we produced here will help to perform such studies. The sWGA method provides a simple and efficient way to study the genomes of *P. ovale curtisi* and *P. ovale wallikeri* mono-infections. We recommend using it in cases where leukodepletion is not applicable, such as low volume of blood sample available, low parasite density, or studies on archived DNA.

## MATERIALS AND METHODS

**Sample collection.** *P. ovale* spp. isolates were selected from the FNMRC database based on parasite densities and the countries where the patients got contaminated.

Genomic DNA was extracted from 200 $\mu$L of whole blood using MagNA Pure automaton (Roche diagnostics, USA) and eluted in 100 $\mu$L. *P. ovale* spp. mono-infection was confirmed with the species-specific quantitative PCR (qPCR) *Plasmodium* typage kit (Bio-Evolution, France) targeting the 18s rRNA for *P. ovale* spp. and the human *beta actin* gene to evaluate human DNA contamination. The reaction was carried out on a ViiA 7 thermocycler (Applied Biosystems). One positive and one negative control were included in each run. $\Delta$Ct (cycle threshold) was defined as the difference between the Ct of *P. ovale* spp. and the human Ct. We performed in-house qPCR high resolution melting (HRM) to differentiate *P. ovale wallikeri* from *P. ovale curtisi* (42).

**Ethical statement.** No specific consent from patients was required since clinical and biological data were collected from the FNMRC database in accordance with the common public health mission of all National Reference Centers in France, in coordination with the Santé Publique France organization for malaria surveillance and care. The study was considered noninterventional research according to article L1221–1.1 of the public health code in France and only requires the nonopposition of the patient (per article L1211–2 of the public health code). All data were anonymized before use. Human DNA was not analyzed.

**Sample filtration.** As a positive control of *P. ovale* spp. high-quality WGS, we adapted to *P. ovale* spp. the parasitize whole blood filtration protocol previously developed by the MalariaGEN consortium for *P. falciparum* infections (https://www.malariagen.net/resources/partner-study-resources/archive-partner-study-resources). This filtration procedure removes the leukocytes carrying human DNA from the infected blood sample. Briefly, 1 g of cellulose powder (MN2100ff cellulose powder, Macherey-Nagel) was transferred into a 10-mL column (BD Emerald syringe 10 mL) and washed with 4 mL of 1× PBS. We slightly modified the filtration protocol and used 200 to 400 $\mu$L red blood cells instead of 2 mL of whole blood to fit our requirements. Blood samples were diluted in 1× PBS to reach a final volume of 2 mL and transferred into the column. A plunger was used to help the blood pass through the column. The column was rinsed three to four times with 4 mL of 1× PBS, and the eluate was centrifuged at 4,000 rpm for 10 min. DNA was then extracted following previously described protocol (see Sample Collection).

**Primer design.** We used the sWGA program available on GitHub (https://github.com/eclarke/swga) to define two sets of primers that bind preferentially to foreground *P. ovale curtisi* (PocGH01, GenBank assembly accession: GCA_900090035.2) or *P. ovale wallikeri* (PowCR01, GenBank assembly accession: GCA_900090025.2) genomes over background human genome (GRCh38, UCSC genome browser) (43). The following sWGA parameters were selected to generate the sets of primers: primers' melt temperature between 18 and 35℃, minimum foreground primers' binding at 414 (equivalent to a binding every 50,000 nucleotides), maximum background primers' binding at 12,837 (equivalent to a binding every 250,000 nucleotides), and a minimum primer size of six nucleotides. We then computationally tested different sets composed of 5 to 10 primers with the following parameters: minimum binding distance between two primers on the background genome of 30,000 nucleotides and maximum binding distance between two primers on the foreground genome of 90,000 nucleotides. Each set is scored using the average distances between primer binding sites on the foreground and background genomes and the Gini index of foreground binding sites (43). The closer the score is to 0, the more specific the primer set is to the parasite genome.

**Selective whole-genome amplification.** The sWGA reactions were performed using one set of primers for *P. ovale curtisi* and one set of primers for *P. ovale wallikeri*. Each reaction was performed in 0.2 mL PCR tubes containing at least 20 ng of template genomic DNA, 0.1 mg/mL bovine serum albumin (BSA) (New England Biolabs), 1 mM dNTPs (New England Biolabs), 2.5 $\mu$M each primer, 1× Phi29 reaction buffer (New England Biolabs), 30 units of Phi29 polymerase (New England Biolabs), and molecular biology-grade water to reach a final reaction volume of 50 $\mu$L. The reaction was carried out on a thermocycler (Mastercycler Gradient, Eppendorf) with the following step-down program: 5 min at 35℃, 10 min at 34℃, 15 min at 33℃, 20 min at 32℃, 25 min at 31℃, 16 h at 30℃, then heating for 15 min at 65℃ to inactivate the Phi29 polymerase before cooling to 4℃. Amplified products were quantified using the Qubit dsDNA high-sensitivity kit (Thermo Fisher Scientific). Amplified products were cleaned using Agencourt Ampure XP beads (Beckman Coulter) as follows: 1.8 volumes of beads were added to 1 volume of amplified products, briefly mixed, and then incubated for 5 min at room temperature. A magnetic rack was used to capture the DNA binding beads. The beads were then washed twice using 200 $\mu$L of 80% ethanol, and DNA was eluted with 60 $\mu$L of EB buffer.

**Methylation digest.** Twenty microliters of genomic DNA extracted from *P. ovale* spp.-infected blood samples were digested before sWGA with 10 units of McrBC (New England Biolabs, United Kingdoms) in a 30-$\mu$L reaction mix containing 1× NEBuffer 2, 0.5 $\mu$L of 100× BSA, and 0,5 $\mu$L of 100× GTP (New England Biolabs, United Kingdoms) as previously performed on *P. falciparum* isolates (24). Samples were incubated for 2 h at 37℃ then inactivated at 65℃ for 20 min according to the manufacturer's recommendations.

**Whole-genome sequencing.** Sequencing libraries were prepared with 250 ng or 50 $\mu$L of DNA for sWGA samples and filtered controls, respectively, using the KAPA HyperPrep Library Preparation kit (Kapa Biosystems, Woburn, MA) following manufacturer's instructions. Mechanical DNA shearing was performed with the Covaris S220 through microTube-50 AFA Fiber Screw-Cap (Covaris) using the following settings: 30% duty factor, 100W peak incidence power, and 1,000 cycles per burst for 150 s. DNA libraries were then checked for quality and quantity using Qubit for concentration and BioAnalyzer 2100 Agilent for fragment size. Libraries were sequenced on an Illumina NextSeq 500 System using 150-bp paired-end sequencing chemistry at the GENOM'IC platform from Institut Cochin (Paris, France).

Raw fastq files were aligned to the PowCR01 or PocGH01 reference genomes using the BWA-mem (Burrows-Wheeler Aligner) algorithm (default parameters) (44). Aligned reads were processed using SAMtools v.1.4 (45). Coverage statistics and depth estimates were obtained using BEDtools v.2.26.0 (46). To confirm the

specificity of the short reads generated, raw reads were aligned to a concatenated genome of *P. falciparum* (Pf3D7, PlasmoDB release 57), *P. malariae* (PmUG01, PlasmoDB release 57), and *P. ovale curtisi/P. ovale wallikeri* (PocGH01 or PowCR01). We finally took fastq from previously published *P. malariae* (ERR4019168) or *P. falciparum* (ERR636035) data and aligned them against the concatenates resistances-associated gene sequences (PF3D7_0417200, PF3D7_0810800, PF3D7_0709000, PF3D7_0523000, PF3D7_1343700, PmUG01_05034700, PmUG01_14045500, PmUG01_01020700, PocGH01_05028400, PocGH01_14036800, PmUG01_10021600, PmUG01_12021200, PocGH01_01016900, PocGH01_10018700, PocGH01_12019400, PowCR01_050023500, PowCR01_140031200, PowCR01_010011800, PowCR0100013900, and PowCR01_120015100).

**Determination of *P. ovale wallikeri* genes orthologous to *P. ovale curtisi*.** To identify *P. ovale wallikeri* protein-coding genes orthologous to *P. ovale curtisi*, we aligned PocGH01 *P. ovale curtisi* proteins (extracted from PlasmoDB release 49) to the *P. ovale wallikeri* proteome (GenBank assembly accession: GCA_900090025.2) with BLAST+ (version 2.11.0) (27). Homologous proteins were identified using two criteria: a significant BLAST E-value of $<10^{-3}$ and a similar protein sequence size (difference length of maximum 20%). We used the OrthoMCL algorithm to confirm that previously identified proteins shared common functions and could therefore be defined as orthologous (28).

**Variant calling and analysis.** Duplicate reads were tagged and removed using Picard MarkDuplicates (v. 2.26.10). Single-nucleotide polymorphisms (SNPs) were identified using BCFtools *mpileup* version 1.13 (minimum mapping quality for an alignment to be used [q]: 20; minimum base quality for a base to be considered [Q]: 20; coefficient for downgrading mapping quality for reads containing excessive mismatches [C]: 50) (47, 48), followed by BCFtools *call* with the option *–V indels* to discard indels. The following filtration quality criteria were used with GATK (v.4.1.8) (49): mapping quality of 40, a Phred-scaled quality score of 30, and a Phred-scaled *P* value using Fisher's exact test to detect strand bias of $>60$. Variant calling analysis was restricted to chromosomes. Positions covered with less than five reads were filtered out. For each isolate, we considered a position as biallelic if a minimum of five reads mapped to both reference and alternative alleles. Following the Pearson's method (32), we calculated and plotted the nonreference allele frequency (NRAF) and the percentage of heterozygote calls across the chromosomes for each isolate for the SNPs obtained with the sWGA + McrBC approach.

**Phylogenetic analysis.** The interspecies phylogenetic tree was reconstructed using the top covered gene of *P. ovale* spp. Gene sequences of our *P. ovale* spp. samples were determined by producing pileup files (containing information on matches, mismatches, indels, strand, and mapping quality) using SAMtools *mpileup* from bam files, then consensus sequences were determined using a Perl homemade script. Nucleotide sequences were then translated into protein sequences, and for each was identified the orthologous to *P. berghei*, *P. chabaudi*, *P. cynomolgi* strain M, *P. falciparum* 3D7 (50), *P. gallinaceum*, *P. knowlesi* strain M, *P. malariae*, *P. reichenowi*, *P. vivax* P01 (51), *P. vivax*-like, and *P. yoelii* parasites. Each set of protein sequences for a given *Plasmodium* was independently aligned using MAFFT v.7.307 (52) with default options, then cleaned using GBlocks v.0.91b (53) to automatically remove noninformative and gapped sites. The cleaned, nonzero length alignments were then concatenated. The maximum likelihood phylogenetic tree was inferred using IQTREE (v. 1.6.12) after determining the best-fitting amino acid exchange rate matrix (54). Branch supports were assessed with 1,000 bootstrap replicates using the ultrafast bootstrap approximation implemented in IQTREE. The phylogenetic tree was manipulated and visualized with treeio and ggtree R packages respectively (55, 56).

**Data analysis.** For all statistical tests, the number of reads was expressed in millions of reads. We normalized the depth of coverage with the overall number of reads in millions (normalized coverage) to compare WGS data sets having different numbers of total reads. Quantitative variables were expressed as median (10th to 90th percentile). Mann-Whitney *U* test was used to compare medians. Proportions were compared using the $\chi^2$ or Fisher's exact tests. The Kolmogorov-Smirnov test was used to assess the normality of variable distributions, and the Levene's test to verify the homogeneity of the variances. When both criteria were validated, we used the Pearson correlation test; otherwise, a Spearman's rank test was performed. All statistical analyses and graphs were performed on R software v. 4.0.3 (57). To visualize read sequence distribution across parasite chromosomes, the average read depth in 2-kb windows across the 14 chromosomes was calculated then displayed with Circos software (58).

**Data availability.** The genome Illumina sequencing reads from the *P. ovale curtisi* and *P. ovale wallikeri* samples produced in the sWGA + McrBC condition were deposited in the European Nucleotide Archive under the accession number PRJEB51041. All scripts used in this study were deposited on GitHub (https://github.com/Rcoppee/P_ovale_sWGA_project).

## SUPPLEMENTAL MATERIAL

Supplemental material is available online only.

**SUPPLEMENTAL FILE 1**, PDF file, 1.6 MB.
**SUPPLEMENTAL FILE 2**, XLSX file, 0.2 MB.
**SUPPLEMENTAL FILE 3**, XLSX file, 0.03 MB.

## ACKNOWLEDGMENTS

We thank all the French National Malaria Reference Center members and correspondents who included the samples.

There are no conflicts of interest to disclose.

V.J. and S.H. designed the study. V.J. performed the experimentations, analyzed the data, and wrote the article. V.J., E.G., and R.C. performed the bioinformatics analyses. All the authors reviewed the article. S.H. supervised the study.

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
