## [Reviewer comments · Microbiology Spectrum]

Microbiology Spectrum

Development and optimization of a selective whole-genome amplification to study *Plasmodium ovale* spp.

Valentin Joste, Emilie Guillochon, Jérôme Clain, Romain Coppée, and Sandrine Houzé

Corresponding Author(s): Valentin Joste, French National Malaria Reference Center

Review Timeline:

Submission Date:	March 3, 2022
Editorial Decision:	June 3, 2022
Revision Received:	July 17, 2022
Editorial Decision:	August 12, 2022
Revision Received:	August 21, 2022
Accepted:	August 25, 2022

Editor: Jennifer Guler

Reviewer(s): The reviewers have opted to remain anonymous.

Transaction Report:

DOI: <https://doi.org/10.1128/spectrum.00726-22>

June 3, 2022

Dr. Valentin Joste
French National Malaria Reference Center
valentinjoste@gmail.com
Paris
France

Re: Spectrum00726-22 (Development and optimisation of a selective whole-genome amplification to study *Plasmodium ovale* spp.)

Dear Dr. Valentin Joste:

Due to limited availability of reviewers, we were only able to secure one formal review. However, I have evaluated the manuscript and have provided some comments and would like these addressed in the resubmission.

-It would be informative to add a read coverage comparison between non-SWGA and SWGA samples. This would facilitate appreciation of the benefit for using the method.

-Do you know the success rate for clinical sample amplification? Did you encounter any samples that failed? This information is useful for others planning to use this method.

-Do you have recommendations about when to use the various methods? For example, are there sample characteristics that necessitate SWGA vs leukodepletion (perhaps blood volume?).

-How many SNPs are called by both methods? How many overlap between the two methods? These metrics are important because you show in Figure 3 that read count is uneven following SWGA.

-Please provide access to the Perl homemade script used in the phylogenetic analysis.

For Supp Fig 3: there are assumptions when using Pearson correlations- it is important to check that your data fits these assumptions and whether outliers should be removed. For example, outliers should not be included for calculating R2.

Minor points:

1. Legend of Supp fig 3: "red points a difference between 50 to 75%" should be "blue points a difference between 50 to 75%"
2. Title: optimisation -> optimization
3. Standardize the format of Supplementary Tables and the name of sheets

Link Not Available

Sincerely,

Jennifer Guler

Journals Department
American Society for Microbiology
1752 N St., NW

Reviewer comments:

Reviewer #1 (Comments for the Author):

The premise of this manuscript is great for malaria biology, where there is limited knowledge on the evolution of other species of human malaria parasites, especially *P. malariae* and *P. ovale*. In this manuscript, the authors present a method to specifically amplify the species of *P. ovale* towards future population genomics analyses that may be informative of its evolution and guide interventions for the complete elimination of all malaria parasites. The method they have presented, though not novel, is applied here to *P. ovale* species for the first time. There are a number of issues that need to be resolved;

1. The authors should elaborate more on how the background from the human genome and contamination with *P. falciparum* was assessed in the design of SWGA primers and their use. The only indication is that the SGWA primers for each species were designed, eliminating the other as background. So, were the primer sets specific per species. Can a gel showing no amplification for *P. falciparum*, humans, and the non-target species be shown? In natural infections, *P. ovale* is mostly seen as a co-infection with *P. falciparum* and sometimes with *P. malariae* too. So eliminating these would help to ensure that these primers can be used in the field.
2. The authors only used suppose mono-infections. Following from the above, these are rare, and larger genomic studies would need to deal with co-infecting Plasmodium species. No evidence on whether the short reads from these mono-infections can map to *falciparum* or malaria. This control against these other species will be clear evidence of specificity.
3. Controls were leukocyte depleted. It is not clear how these were chosen to be controls. Was SWGA also applied to these controls? To determine the effectiveness of SWGA, SWGA and non-SWGA sequences from the same sample should be compared.
4. For others to use this protocol, it will be helpful to know how much DNA from controls and SWGA was used for library prep, in case these were not amplified. Was the sequencing library prep PCR free or PCR based
5. It is not clear if the genome coverage report in the main text is for all samples combined and if so, is 10x the mean of median coverage. Did this included coverage for the controls as well
6. From the scatter plot of parasitemia vs difference between SWGA and McrBc-SWGA, sample IDs could help with clarity
7. Considering that it is not clear if the short reads generated were mapped against *P. falciparum* orthologues of drug resistance genes, it is possible that any co-sequenced *P. falciparum* or *P. malariae* drug resistance targets would result in variants. As real-time PCR seems to indicate that these were mono-infections, the authors may have to discuss how this will be applied for wild isolates with contaminating co-infections
8. For the number of samples sequenced, true allele frequencies cannot be determined. If the frequencies reported were from vcf-tools, then the authors need to indicate that these were determined from read counts and not from consensus data.
9. For the total number of variants detected, the numbers are not clear. For example, 9,782 (3,326 per sample). This seems to be for 3 samples rather than the 5 samples sequenced.
10. Considering there are non-chromosomal contigs for *P. ovale* species, why were these not also used to map reads. Alternatively, the authors could enrich the manuscript by attempting de-novo assembly.
11. For the interest of the Plasmodium genomics community, genome-wide plots of heterozygosity would be informative, though this will be limited given the sample size.

Overall, the work presented has merits and can be improved. The discussion winds through a repeat of the results, rather than contextualising the outcomes of the work.

Staff Comments:

Preparing Revision Guidelines

Please return the manuscript within 60 days; if you cannot complete the modification within this time period, please contact me. If you do not wish to modify the manuscript and prefer to submit it to another journal, please notify me of your decision immediately so that the manuscript may be formally withdrawn from consideration by Microbiology Spectrum.

Dear Dr. Valentin Joste:

Due to limited availability of reviewers, we were only able to secure one formal review. However, I have evaluated the manuscript and have provided some comments and would like these addressed in the resubmission.

-It would be informative to add a read coverage comparison between non-SWGA and SWGA samples. This would facilitate appreciation of the benefit for using the method.

Thanks for this suggestion. We did not perform non-sWGA sequencing to prove the benefits of using sWGA. In fact, several publications have already published data about the failure of next generation sequencing without preamplification or filtration (1–4). In the figure below, published by Oyola *et al* (1), sequencing of *Plasmodium falciparum* clinical isolates without selective preamplification (corresponding to WGA in the figure) is not possible, with less than 5% of reads that mapped to *P. falciparum*.

Fig. 4 Selective whole genome amplification (sWGA) enrichment. Simulated clinical samples comprising 96% human DNA and 4% *P. falciparum* DNA (3D7) were amplified using either WGA or sWGA. Amplified samples were sequenced to determine the proportion of reads mapping to human or *P. falciparum* reference genomes

Figure 1 – sWGA enrichment. from Oyola et al (1).

We considered as unnecessary to test the non-sWGA condition based on those previous observations. On the contrary, we compared the read coverage of the sWGA and leukodepleted samples (considered as controls) to validate the use of sWGA (see figure S3).

-Do you know the success rate for clinical sample amplification? Did you encounter any samples that failed? This information is useful for others planning to use this method.

We did not observe any failure in clinical sample amplification. But we did not test samples with parasite density below 1,790 parasites/ μ L for *P. ovale curtisi* and 198 p/ μ L for *P. ovale wallikeri* and we cannot predict the success of the amplification below those levels. This is now clearly state in the discussion lines 405.

-Do you have recommendations about when to use the various methods? For example, are there sample characteristics that necessitate SWGA vs leukodepletion (perhaps blood volume?).

We consider that the best overall technique to amplify *P. ovale* spp clinical samples remains the leukodepletion because it provides homogeneity in reads mapping compared to sWGA. Therefore, leukodepletion allows read distribution-based analyses such as the measure of gene copy number variation (5), related to some drug resistance in *Plasmodium* (6).

In our lab, we used sWGA for samples that we could not filter in those situations:

- low volume of blood available (minimal red blood cells volume of 200 μ L for leukodepletion);
- retrospective study before the implementation of leukodepletion on fresh blood samples;

We add those recommendations in the discussion, line 461 to 463.

-How many SNPs are called by both methods? How many overlap between the two methods? These metrics are important because you show in Figure 3 that read count is uneven following SWGA.

For Poc1, we respectively called 3,732 and 6,980 SNPs with the sWGA and the filtered approaches. 3,638 SNPs overlapped with both methods.

97.4% of SNPs called in sWGA overlap with those from the filtered sample with only 94 SNPs not counted in filtration (64 that did not fit our quality requirement, 8 unmutated positions (0 in coding regions, one NRAF > 0,5 in the sWGA condition) and 22 SNPs with insufficient coverage). On the other side, 52.1% of the SNPs called with the filtration were found in sWGA. The remaining 3,228 positions not detected in sWGA were as follow:

- 1,283 positions that did not fit our quality requirement,
- 1,594 positions with insufficient coverage (but still mainly 0/1 or 1/1) and 409 positions with no coverage,
- 56 unmutated positions (25 in coding regions, NRAF always < 0,5 in filtered condition).

For Pow1, we respectively called 6,045 and 6,125 SNPs with the sWGA and the filtered approaches. 4,793 SNPs overlapped with both methods.

79.3% of SNPs called in sWGA overlap with those from the filtered sample. The remaining 1,252 positions were as follows: 725 positions that did not fit our quality requirement, 394 positions with insufficient coverage and 5 with no coverage, and 128 unmutated positions (49 in coding regions, 1 with NRAF > 0,5).

On the other side, 78.3% of SNPs called with the filtration were found in sWGA. The remaining 1,332 positions were as follows: 696 positions that did not fit our quality requirement, 440 positions with insufficient coverage, 117 positions with no coverage and 79 unmutated positions (18 in coding regions, 2 with NRAF > 0,5).

Please look at figure 2 downside for more details. We add the figure but not the detailed data in the new submitted files and lines 368 to 370 (Figure S6).

Figure 2– Comparison of the SNPs obtained with the sWGA and the filtration methods for both *P. ovale curtisi* (A), *P. ovale wallikeri* (B) and NRAF of the positions with different genotypes between sWGA and filtration (0/0 to 0/1 or 1/1) (C). Overlap category correspond to identical SNPs obtained with both methods. Quality issue correspond to SNPs obtained for one method that did not pass the quality filters for the other method. Coverage issue correspond to SNPs obtained for one method that did not pass the depth filters for the other method. Unmutated positions correspond to positions with SNPs (0/1 or 1/1) for one method and no mutation (0/0) with the other method. Poc stands for *P. ovale curtisi* and Pow stands for *P. ovale wallikeri*.

-Please provide access to the Perl homemade script used in the phylogenetic analysis.

You will find the Perl script following this link: https://github.com/Rcoppee/P_ovale_sWGA_project. The link has been added in the main text line 481-482.

For Supp Fig 3: there are assumptions when using Pearson correlations- it is important to check that your data fits these assumptions and whether outliers should be removed. For example, outliers should not be included for calculating R².

You are right, our data did not fit the assumptions of Pearson correlations test after Kolmogorov-Smirnov and Levene test use. We then performed the Spearman rank test to compare the NRAF of both methods.

We ponder the use of R² and decided to remove it. In fact, we did not try to establish a mathematical link between the NRAF in sWGA and leukodepletion but only to know if the NRAF (when NRAF > 0) of both methods are correlated.

We completed the statistical part of the method lines 252 to 263.

Minor points:

1. Legend of Supp fig 3: "red points a difference between 50 to 75%" should be "blue points a difference between 50 to 75%"

Thanks for that remark, the modification has been made in the Figure S5.

2. Title: optimisation -> optimization

Thanks for that remark, the modification has been made.

3. Standardize the format of Supplementary Tables and the name of sheets

All supplementary tables have been submitted in one file call Supplemental material except for table S4 and S5 that are too large.

Reviewer comments:

Reviewer #1 (Comments for the Author):

The premise of this manuscript is great for malaria biology, where there is limited knowledge on the evolution of other species of human malaria parasites, especially *P. malariae* and *P. ovale*. In this manuscript, the authors present a method to specifically amplify the species of *P. ovale* towards future population genomics analyses that may be informative of its evolution and guide interventions for the complete elimination of all malaria parasites. The method they have presented, though not novel, is applied here to *P. ovale* species for the first time. There are a number of issues that need to be resolved;

1. The authors should elaborate more on how the background from the human genome and contamination with *P. falciparum* was assessed in the design of SWGA primers and their use. The only indication is that the SGWA primers for each species were designed, eliminating the other as background. So, were the primer sets specific per species. Can a gel showing no amplification for *P. falciparum*, humans, and the non-target species be shown? In natural infections, *P. ovale* is mostly seen as a co-infection with *P. falciparum* and sometimes with *P. malariae* too. So eliminating these would help to ensure that these primers can be used in the field.

When designing primers with the sWGA software, the algorithm asks for a background genome. In our case of *Plasmodium* blood infection, we chose the human genome as background. We did not provide another *Plasmodium* genome as background (such as *P. falciparum*) because only one background genome could be use. The primers' sets were then not designed as specific to *P. ovale* spp over the other *Plasmodium* species but as specific over the human genome. We considered the main issue was the human DNA contamination and not the possible co-infecting *Plasmodium* species (although this could indeed be another issue).

As you can see in the table below (and added in the new submission, table S2), we calculated the number of binding sites of each primer set on different *Plasmodium* species as well as the median distance between two primer binding sites.

	Genome size (s)	Poc primers' set		Pow primers' set	
		Number of binding sites (n)	s/n (bp)	Number of binding sites (n)	s/n (bp)
P. ovale spp	33 Mbp	4,551	7,251	4,986	6,618
P. falciparum 3D7	23 Mbp	2,127	10,814	2,267	10,146
P. malariae UG01	34 Mbp	4,218	8,061	4,791	7,097

P. vivax P01	29 Mbp	2,760	10,507	2,790	10,394
Human	2948 Mbp	23,005	128,151	25,492	115,644

Table 1 - Number of primers' binding sites (n) and the ratio of genome size/number of binding site (s/n) for the PocGH01 or PowCR01, Pf3D7, PmUG01, PvP01 and GRCh38 genomes for each Poc and Pow primers' sets.

The number of primers' binding is twice larger for *P. ovale wallikeri* or *P. ovale curtisi* compared to *P. falciparum*. Besides, the ratio s/n is twice higher for *P. falciparum* than *P. ovale* spp genome. The primers will preferentially bind *P. ovale* spp DNA over *P. falciparum* but I'm not sure it will be sufficient when *P. falciparum* had much higher parasitaemia than *P. ovale* spp. In our experience in imported malaria with qPCR data and as previously published (7), *P. falciparum* parasite density is in most cases largely higher than *P. ovale* spp parasite density (median Ct [IQR] of *P. falciparum* = 21,5 [18-31]; median Ct of *P. ovale* spp = 34,5 [29-37]; p=0.003, Mann-Whitney U-test. See (8) for the detailed target of the qPCR).

I have added a paragraph about the sWGA limitations in the discussion lines 441 to 452, notably for *Plasmodium* co-infections. One of the solution to overcome this issue could be hybridization capture methods (9, 10).

2. The authors only used suppose mono-infections. Following from the above, these are rare, and larger genomic studies would need to deal with co-infecting *Plasmodium* species. No evidence on whether the short reads from these mono-infections can map to *falciparum* or malaria. This control against these other species will be clear evidence of specificity.

You are right. To overcome this hypothesis, we concatenate *P. ovale curtisi* or *P. ovale wallikeri* genome with *P. falciparum* genome (Pf3D7, PlasmoDB release 57) and *P. malariae* genome (PmUG01, PlasmoDB release 57) and aligned *P. ovale* spp sequencing reads against this new reference genome. As presented as an example on the plot below, *P. ovale curtisi* (Figure 2A) or *P. ovale wallikeri* (Figure 2C) reads mapped in large majority to *P. ovale* spp genome and not to *P. falciparum* or *P. malariae* genomes. Besides, reads that mapped to *P. falciparum* or *P. malariae* genomes were of poorest quality (Figure 2B and 2D) and of lowest insert size (figure 3A and 3B). Figure 2A to 2D were added to the new submission (figure S2).

Figure 2 – Coverage and mapping quality of *P. ovale curtisi* (Poc1, A and B) and *P. ovale wallikeri* (Pow1, C and D) short reads generated by swGA against a concatenate genome of *P. malariae* (PmUG01, LT594622 to LT594635), *P. falciparum* (PF3D7, PF3D7_01_v3 to PF3D7_14_v3) and *P. ovale curtisi/P. ovale wallikeri* (PocGH01 or PowCR01, LT594582 to LT594595 or LT594505 to LT594518). Plots were generated using Qualimap (v2.2.1)(11).

The insert sizes displayed in the figure 3 represent the part of the paired-end reads that mapped to the reference. The lowest insert size that mapped to *P. falciparum* or *P. malariae* genome (~20 bp, see figure 4) probably represents short consensus sequences between *Plasmodium* species.

Figure 3 – Insert size across reference for A) *P. ovale wallikeri* and B) *P. ovale curtisi*. Plots were generated using Qualimap (11).

Figure 4 – Comparison of *P. ovale curtisi* reads alignment to *P. falciparum* (on the left) or *P. ovale curtisi* (on the right). Gray bases correspond to identical bases as the reference genome. Other colored bases correspond to bases different to the reference genome. Soft clipped bases are bases that are not part of the alignment because of non-identity with the reference genome. Image from IGV (version 2.8.13).

Figure 3 and figure 4 were not submitted with the manuscript and were only made for the response to the reviewers.

3. Controls were leukocyte depleted. It is not clear how these were chosen to be controls. Was SWGA also applied to these controls? To determine the effectiveness of SWGA, SWGA and non-SWGA sequences from the same sample should be compared.

We prospectively chose one *P. ovale curtisi* and one *P. ovale wallikeri* samples received in the French National Malaria Reference Center to be filtered (Poc1 and Pow1). We applied the sWGA to these leukodepleted controls (see table 1, figure 1B) to evaluate the effectiveness of sWGA. We compared the SNPs obtained by the two methods and obtained really closed NRAF (see figure S5).

We did not perform non-sWGA (without leukodepletion) sequencing to prove the benefits of using sWGA. In fact, several publications have already published data about the failure of *Plasmodium* sequencing without preamplification or filtration (1–4). We considered as unnecessary to test the non-sWGA condition based on those previous observations.

4. For others to use this protocol, it will be helpful to know how much DNA from controls and SWGA was used for library prep, in case these were not amplified. Was the sequencing library prep PCR free or PCR based

For the library, 250 ng of DNA was used when possible. For leukodepleted controls, DNA concentration was very low (0,452 ng/μL for Pow1 and 0,114 ng/μL for Poc1) and 50 μL was used (22,6 ng for Pow1 and 5,7 ng). We add those details in the methods section line 187. The sequencing library protocol was PCR-based.

5. It is not clear if the genome coverage report in the main text is for all samples combined and if so, is 10x the mean of median coverage. Did this included coverage for the controls as well

The mean coverage reported in the main text (32x for *P. ovale curtisi* and 24x for *P. ovale wallikeri*) is for the sWGA method. For the sWGA + McrBC approach, the mean coverage is 62x for *P. ovale curtisi* and 83x for *P. ovale wallikeri*. Those results do not include the leukodepleted controls (93x for Poc1 and 99x for Pow1).

We also compared between the different methods the percentage of the genome covered with at least 10x (see figure S3b).

6. From the scatter plot of parasitemia vs difference between SWGA and McrBc-SWGA, sample IDs could help with clarity

We modified the figure with samples IDs.

7. Considering that it is not clear if the short reads generated were mapped against *P. falciparum* orthologues of drug resistance genes, it is possible that any co-sequenced *P. falciparum* or *P. malariae* drug resistance targets would result in variants. As real-time PCR seems to indicate that these were mono-infections, the authors may have to discuss how this will be applied for wild isolates with contaminating coinfections

Concatenating sequences with *P. ovale curtisi* or *P. ovale wallikeri*, *P. falciparum* and *P. malariae* resistance genes sequences clearly help to eliminate this hypothesis.

We took fastq from previously published *P. malariae* (ERR4019168 (12)) or *P. falciparum* (ERR636035 (1)) data and aligned them against the concatenated resistances associated-genes sequences (PF3D7_0417200, PF3D7_0810800, PF3D7_0709000, PF3D7_0523000, PF3D7_1343700, PmUG01_05034700, PmUG01_14045500, PmUG01_01020700, PocGH01_05028400, PocGH01_14036800, PmUG01_10021600, PmUG01_12021200, PocGH01_01016900, PocGH01_10018700 and PocGH01_12019400).

- | | |
|----------------------------|-----------------------------|
| ① PmUG01_05034700: dhfr-ts | ① PocGH01_05028400: dhfr-ts |
| ② PmUG01_1404500: dhps | ② PocGH01_14036800: dhps |
| ③ PmUG01_01020700: crt | ③ PocGH01_01016900: crt |
| ④ PmUG01_10021600: mdr1 | ④ PocGH01_10018700: mdr1 |
| ⑤ PmUG01_12021200: k13 | ⑤ PocGH01_12019400: k13 |

Figure 5 – Coverage across reference and insert size across reference for ERR4019168 on five genes of *P. malariae* or *P. ovale curtisi* orthologous of known resistance genes. Plots were generated using Qualimap (11).

- | | |
|--------------------------|-----------------------------|
| ① Pf3D7_0417200: dhfr-ts | ① PocGH01_05028400: dhfr-ts |
| ② Pf3D7_0810800: dhps | ② PocGH01_14036800: dhps |
| ③ Pf3D7_0709000: crt | ③ PocGH01_01016900: crt |
| ④ Pf3D7_0523000: mdr1 | ④ PocGH01_10018700: mdr1 |
| ⑤ Pf3D7_1343700: k13 | ⑤ PocGH01_12019400: k13 |

Figure 6 - Coverage across reference and insert size across reference for ERR636035 on five genes of *P. falciparum* or *P. ovale curtisi* orthologous of known resistance genes. Plots were generated using Qualimap (11).

As presented in the figure 5 (for *P. malariae*) and figure 6 (for *P. falciparum*), no *P. ovale curtisi* genes were covered with *P. malariae* or *P. falciparum* reads.

Figures 5 and 6 were not added to the new submission.

8. For the number of samples sequenced, true allele frequencies cannot be determined. If the frequencies reported were from vcftools, then the authors need to indicate that these were determined from read counts and not from consensus data.

I do not calculate allelic frequencies in this study.

9. For the total number of variants detected, the numbers are not clear. For example, 9,782 (3,326 per sample). This seems to be for 3 samples rather than the 5 samples sequenced.

The total number of variants detected is the total of unique SNPs detected, not the sum of the SNPs detected in each sample. If two samples displayed the same SNP, it counts for one and not two SNP. We rephrase the sentences for clarification line 362.

10. Considering there are non-chromosomal contigs for *P. ovale* species, why were these not also used to map reads. Alternatively, the authors could enrich the manuscript by attempting de-novo assembly.

We actually mapped the reads against chromosomes and contigs of the reference genome and the mapping data we presented are against the whole genome. But we only used the reconstructed chromosomes for SNPs calling because the contigs are mainly composed of *pir* gene of *P. ovale* spp. Those *pir* genes are highly variables (such as *var* genes in *P. falciparum*) and is the largest *Plasmodium* multigene family (13). Due to their high variability among clinical *Plasmodium* isolates, alignment is not sufficient to obtain high quality data. Local reconstructions, such as previously published for *var* genes (14), are necessary. No script is actually available to easily reconstruct *P. ovale* spp *pir* genes and we decided to not analyze the SNPs obtained on those contigs.

We agree that *de-novo* assembly would have been of great interest to improve the data. Unfortunately, we do not have the capacity of performing such analysis in our lab.

11. For the interest of the Plasmodium genomics community, genome-wide plots of heterozygosity would be informative, though this will be limited given the sample size.

We add the NRAF profiles as well as the genome-wide plots of the *P. ovale* spp isolates in the figure S4a and S4b as previously done by Pearson et al (15). Percentage of heterozygote calls were low for both species (<0,1%). We saw less heterozygosity for *P. ovale curtisi*, maybe linked to the lower depth of coverage compared to *P. ovale wallikeri* (figure S3 in the Supplemental material).

NRAF plots were quite difficult to interpret in the absence of any reference for *P. ovale* spp. The heterozygous SNPs we see may be related to background noises due to imperfect reconstructed reference genome or misalignment of reads due to paralogous regions (15).

More genomic data are needed to:

- improve the reference genome,
- compute hyperheterozygosity score and used it in the variant filtering approach such as described for *P. vivax* (15).

12. Overall, the work presented has merits and can be improved. The discussion winds through a repeat of the results, rather than contextualizing the outcomes of the work.

We modified the discussion.

Bibliography

1. Oyola SO, Ariani CV, Hamilton WL, Kekre M, Amenga-Etego LN, Ghansah A, Rutledge GG, Redmond S, Manske M, Jyothi D, Jacob CG, Otto TD, Rockett K, Newbold CI, Berriman M, Kwiatkowski DP. 2016. Whole genome sequencing of *Plasmodium falciparum* from dried blood spots using selective whole genome amplification. *Malar J* 15:597.
2. Benavente ED, Gomes AR, De Silva JR, Grigg M, Walker H, Barber BE, William T, Yeo TW, de Sessions PF, Ramaprasad A, Ibrahim A, Charleston J, Hibberd ML, Pain A, Moon RW, Auburn S, Ling LY, Anstey NM, Clark TG, Campino S. 2019. Whole genome sequencing of amplified *Plasmodium knowlesi* DNA from unprocessed blood reveals genetic exchange events between Malaysian Peninsular and Borneo subpopulations. *Scientific Reports* 9:9873.
3. Ibrahim A, Diez Benavente E, Nolder D, Proux S, Higgins M, Muwanguzi J, Gomez Gonzalez PJ, Fuehrer H-P, Roper C, Nosten F, Sutherland C, Clark TG, Campino S. 2020. Selective whole genome amplification of *Plasmodium malariae* DNA from clinical samples reveals insights into population structure. *Scientific Reports* 10:10832.
4. Cowell AN, Loy DE, Sundararaman SA, Valdivia H, Fisch K, Lescano AG, Baldeviano GC, Durand S, Gerbasi V, Sutherland CJ, Nolder D, Vinetz JM, Hahn BH, Winzeler EA. 2017. Selective Whole-Genome Amplification Is a Robust Method That Enables Scalable Whole-Genome Sequencing of *Plasmodium vivax* from Unprocessed Clinical Samples. *mBio* 8.
5. Beghain J, Langlois A-C, Legrand E, Grange L, Khim N, Witkowski B, Duru V, Ma L, Bouchier C, Ménard D, Paul RE, Ariey F. 2016. *Plasmodium* copy number variation scan: gene copy numbers evaluation in haploid genomes. *Malar J* 15:206.
6. Costa GL, Amaral LC, Fontes CJF, Carvalho LH, de Brito CFA, de Sousa TN. 2017. Assessment of copy number variation in genes related to drug resistance in *Plasmodium vivax* and *Plasmodium falciparum* isolates from the Brazilian Amazon and a systematic review of the literature. *Malar J* 16:152.
7. Groger M, Lutete GT, Mombo-Ngoma G, Ntamabyaliro NY, Mesia GK, Mujobu TBM, Mbadanga LBD, Manego RZ, Egger-Adam D, Borghini-Fuhrer I, Shin J, Miller R, Arbe-Barnes S, Duparc S, Ramharther M. 2022. Effectiveness of pyronaridine-artesunate against *Plasmodium malariae*, *Plasmodium ovale* spp, and mixed-*Plasmodium* infections: a post-hoc analysis of the CANTAM-Pyramax trial. *The Lancet Microbe* 0.
8. Schindler T, Robaina T, Sax J, Bieri JR, Mpina M, Gondwe L, Acuche L, Garcia G, Cortes C, Maas C, Daubenberger C. 2019. Molecular monitoring of the diversity of human pathogenic malaria species in blood donations on Bioko Island, Equatorial Guinea. *Malaria Journal* 18:9.
9. Melnikov A, Galinsky K, Rogov P, Fennell T, Van Tyne D, Russ C, Daniels R, Barnes KG, Bochicchio J, Ndiaye D, Sene PD, Wirth DF, Nusbaum C, Volkman SK, Birren BW, Gnirke A, Neafsey DE. 2011. Hybrid selection for sequencing pathogen genomes from clinical samples. *Genome Biol* 12:R73.
10. Bright AT, Tewhey R, Abeles S, Chuquiyaui R, Llanos-Cuentas A, Ferreira MU, Schork NJ, Vinetz JM, Winzeler EA. 2012. Whole genome sequencing analysis of *Plasmodium vivax* using whole genome capture. *BMC Genomics* 13:262.
11. Okonechnikov K, Conesa A, García-Alcalde F. 2016. Qualimap 2: advanced multi-sample quality control for high-throughput sequencing data. *Bioinformatics* 32:292–294.
12. Plenderleith LJ, Liu W, Li Y, Loy DE, Mollison E, Connell J, Ayoub A, Esteban A, Peeters M, Sanz CM, Morgan DB, Wolfe ND, Ulrich M, Sachse A, Calvignac-Spencer S, Leendertz FH, Shaw GM, Hahn BH, Sharp PM. 2022. Zoonotic origin of the human malaria parasite *Plasmodium malariae* from African apes. *Nat Commun* 13:1868.
13. Little TS, Cunningham DA, Vandomme A, Lopez CT, Amis S, Alder C, Addy JWG,

McLaughlin S, Hosking C, Christophides G, Reid AJ, Langhorne J. 2021. Analysis of *pir* gene expression across the Plasmodium life cycle. *Malaria Journal* 20:445.

14. Gq T-H, L T, R N, Hht N, Bf S, Da L, J M, Sa C, Js R, Mj M, Sj R, Gv B, Kp D, Rn P, Nm A, At P, Mf D. 2018. The Plasmodium falciparum transcriptome in severe malaria reveals altered expression of genes involved in important processes including surface antigen-encoding var genes. *PLoS biology* 16.

15. Pearson RD, Amato R, Auburn S, Miotto O, Almagro-Garcia J, Amaratunga C, Suon S, Mao S, Noviyanti R, Trimarsanto H, Marfurt J, Anstey NM, William T, Boni MF, Dolecek C, Tran HT, White NJ, Michon P, Siba P, Tavul L, Harrison G, Barry A, Mueller I, Ferreira MU, Karunaweera N, Randrianarivelojosia M, Gao Q, Hubbart C, Hart L, Jeffery B, Drury E, Mead D, Kekre M, Campino S, Manske M, Cornelius VJ, MacInnis B, Rockett KA, Miles A, Rayner JC, Fairhurst RM, Nosten F, Price RN, Kwiatkowski DP. 2016. Genomic analysis of local variation and recent evolution in Plasmodium vivax. *Nat Genet* 48:959–964.

August 12, 2022

Dr. Valentin Joste
French National Malaria Reference Center
valentinjoste@gmail.com
Paris
France

Re: Spectrum00726-22R1 (Development and optimisation of a selective whole-genome amplification to study *Plasmodium ovale* spp.)

Dear Dr. Valentin Joste:

We appreciate that the authors addressed all of the reviewer comments in the revised manuscript. Overall- all concerns have been addressed. I am recommending to accept the manuscript, however the data on the cross-mapping of reads at resistance loci (between Pf-Po and Pm-Po) that was included in the response to the reviewers needs to be included and referenced in the final version of the manuscript. This data impacts the use of the method with mixed infection samples, an important application of the method.

As you will see your paper is very close to acceptance. Please modify the manuscript along the lines I have recommended. As these revisions are quite minor, I expect that you should be able to turn in the revised paper in less than 2 weeks, if not sooner.

When submitting the revised version of your paper, please provide (1) point-by-point responses to the issues I raised in your cover letter, and (2) a PDF file that indicates the changes from the original submission (by highlighting or underlining the changes) as file type "Marked Up Manuscript - For Review Only". Please use this link to submit your revised manuscript. Detailed instructions on submitting your revised paper are below.

Link Not Available

Sincerely,

Jennifer Guler

Preparing Revision Guidelines

- point-by-point responses to the issues I raised in your cover letter
- Upload a compare copy of the manuscript (without figures) as a "Marked-Up Manuscript" file.
- Each figure must be uploaded as a separate file, and any multipanel figures must be assembled into one file.
- Manuscript: A .DOC version of the revised manuscript
- Figures: Editable, high-resolution, individual figure files are required at revision, TIFF or EPS files are preferred

Please return the manuscript within 60 days; if you cannot complete the modification within this time period, please contact me. If you do not wish to modify the manuscript and prefer to submit it to another journal, please notify me of your decision immediately so that the manuscript may be formally withdrawn from consideration by Microbiology Spectrum.

Dear Dr. Valentin Joste:

We appreciate that the authors addressed all of the reviewer comments in the revised manuscript. Overall- all concerns have been addressed. I am recommending to accept the manuscript, however the data on the cross-mapping of reads at resistance loci (between Pf-Po and Pm-Po) that was included in the response to the reviewers needs to be included and referenced in the final version of the manuscript. This data impacts the use of the method with mixed infection samples, an important application of the method.

As you will see your paper is very close to acceptance. Please modify the manuscript along the lines I have recommended. As these revisions are quite minor, I expect that you should be able to turn in the revised paper in less than 2 weeks, if not sooner.

When submitting the revised version of your paper, please provide (1) point-by-point responses to the issues I raised in your cover letter, and (2) a PDF file that indicates the changes from the original submission (by highlighting or underlining the changes) as file type "Marked Up Manuscript - For Review Only". Please use this link to submit your revised manuscript. Detailed instructions on submitting your revised paper are below.

We added the Figure S8 on the cross-mapping of reads at resistance loci between *P. falciparum* and *P. ovale curtisi*/*P. ovale wallikeri* on one side and between *P. malariae* and *P. ovale curtisi*/*P. ovale wallikeri* on the other side. We modified consequently the methods section lines 203 to 211 and the results section lines 400 to 403.

August 25, 2022

Dr. Valentin Joste
French National Malaria Reference Center
valentinjoste@gmail.com
Paris
France

Re: Spectrum00726-22R2 (Development and optimization of a selective whole-genome amplification to study *Plasmodium ovale* spp.)

Dear Dr. Valentin Joste:

Your manuscript has been accepted, and I am forwarding it to the ASM Journals Department for publication. You will be notified when your proofs are ready to be viewed.

Sincerely,

Jennifer Guler
Editor, Microbiology Spectrum

Journals Department
Supplemental file 3: Accept
Supplemental file 2: Accept
Supplemental Material: Accept